# A New Entropic Measure for the Causality of the Financial Time Series

Peter B. Lerner [1,2]

1  SUNY-Brockport, Brockport, NY 14420, USA; pblerner18@gmail.com or pblerner@syr.edu
2  School of Business Administration, Anglo-American University, ul. Letnikow, 120 Prague, Czech Republic

**Abstract:** A new econometric methodology based on deep learning is proposed for determining the causality of the financial time series. This method is applied to the imbalances in daily transactions in individual stocks and also in exchange-traded funds (ETFs) with a nanosecond time stamp. Based on our method, we conclude that transaction imbalances of ETFs alone are more informative than transaction imbalances in the entire market despite the domination of single-issue stocks in imbalance messages.

**Keywords:** causality; market microstructure; market imbalance; TAQ-ARCA; C-GAN neural network

## 1. Introduction

A conventional method for determining the causality of the financial time series was developed by Clive Granger in the 1980s, who was awarded the Nobel Memorial Prize in 2003. The essence of the method is that a subset of an information set is excluded from analysis, and probability distributions are evaluated on a smaller information set (Diebold 2006; Diks and Panchenko 2006). The generation of probability distributions usually requires fitting the vector autoregressive model (VAR) to the time series and excluding some of the explanatory variables.

Nonparametric versions of the Granger causality tests were developed later, especially in the papers by Baek and Brock (1992), and extended by many authors (Diks and Panchenko 2006, and op. cit.). The Baek and Brock test and its variants are computationally very intensive for the large datasets prevailing in modern securities studies. They require state space coverage with cells of the size $\varepsilon > 0$ and computing correlations between the cells for the decreasing epsilon.

Since the 2010s, transactions in the stock market began to carry nanosecond time stamps. This change requires new methods of analysis adapted to the new realities.[1] The emergence of the big data framework and attempts to use deep learning methods created the following challenge: Regressions became nonlinear, and may contain hundreds of thousands of parameters in the case of this paper—and trillions in the case of Google datasets. Furthermore, deep learning algorithms usually present a "black box", and it is hard to attribute the input changes to the output differences.

The capacity of the human mind to analyze multidimensional time series consisting of billions of market events has remained largely unchanged. Because of our evolution in three-dimensional space, humans have the best grasp of two-dimensional information. Consequently, the methods of image analysis are among the best developed in the whole discipline of signal processing.

My paper adapts deep learning methods developed for image processing to the causality of the financial time series. Comparing two datasets, the one which requires more information to produce a deepfake using a neural network is considered more informative. A precise formulation of these criteria is provided in Section 4.

C-GANs (convolutional generational adversarial neural networks) appeared in 2015. The original purpose of the method was the image analysis and/or generation of deepfakes

(Goodfellow 2017). The essence of the C-GAN is that the network is divided into two parts: generator and discriminator, or critic (Rivas 2020). The generator net produces fake images from random noise and learns to improve them with respect to the training file. The discriminator tries to distinguish fake from real images based on training statistics.

To demonstrate this method's utility, I use it to analyze trading imbalances in New York Stock Exchange (NYSE) trading, which the Securities and Exchange Commission (SEC) requires to be stored with a nanosecond time stamp. These images, for different days, are standardized to ensure their comparability. The imbalance events constitute a situation when the counterparty does not instantly deliver the stock to close its position. The number of these events per day is several million. The time series are preprocessed into two-dimensional images of realistic size to be analyzed using a PC.

Why is this problem important? Given the instances of "flash crashes" in the market, the first and largest of those reported being the Flash Crash of 2010 on the NYSE, the question of whether exchange-traded funds stabilize or destabilize the market became increasingly important. In particular, the Flash Crash was attributed to the toxicity of liquidity of S&P minis orders (Easley et al. 2013). Because of the explosive growth in ETF markets (more information in Section 2), the traded volume in the compound portfolios representing ETF shares can easily exceed trading volume in the underlying securities. Intuitively, this can cause a problem in the delivery of the underlying, which can propagate through the system and, in rare cases, cause a crash. Alternatively, the Mini-Flash crash of 24 August 2015 demonstrated a significant deviation in market index price—subject to the circuit breaking several times—and weighted ETF prices (for a detailed description, see Moise 2023, especially Figure 1).

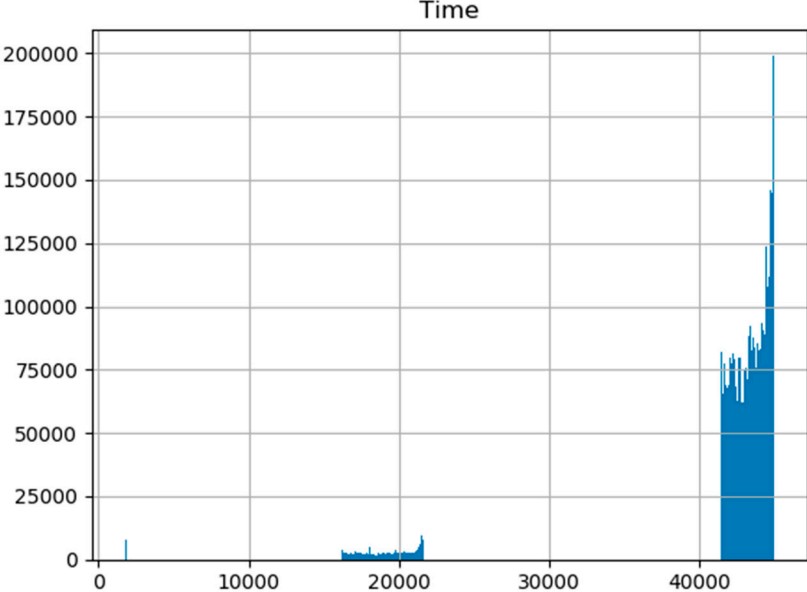

**Figure 1.** The rate of daily imbalance messages in the TAQ ARCA database. During 45,000 s of the day-to-day operation of the system, there were around 4 million messages, the maximum coming at or around 4:00 p.m. The maximum rate during a 100 s interval typically reached 200,000. Most events are concentrated at the beginning, noontime, and the end of the day.

In the current paper, I explore whether the market imbalance events drive ETF imbalances or vice versa. This problem is a good testbed for the proposed deep learning methodology.

The paper is structured as follows. In Section 2, I briefly outline commonly referred information on the ETF market. In Section 3, I describe the structure of the database. Section 4 describes preprocessing of SEC data into two-dimensional images. In Section 5, we establish the causality (information link) between ETF and market transaction imbalances.

Section 6 has an additional robustness check of the proposed results. Section 7 is the summary of the results. The Appendix A describes different kinds of imbalance messages according to NYSE. I review a possible theoretical justification for ETF informational dominance in Appendix B.

## 2. Formulation of the Problem and the Literature Review

The ETF market has exhibited explosive growth in recent years (NYSE 2021). Invented to securitize stock inventories of large asset managers, it developed into a major asset class in its own right (Gastineau 2010). Different anomalies in the markets, particularly the Flash Crash in May 2010, were partly attributed to the instability of exchange-traded products (ETPs), especially the S&P minis (Easley et al. 2013).

The question of whether the introduction of ETFs increase or decrease market quality has been discussed in many authoritative publications (Bhattacharya and O'Hara 2018, and op. cit.), Israeli et al. (2017) mentioned that there are two main opposing reasons for expectations of market quality change. On the positive side, exposure to ETFs provides more information on the underlying stocks, particularly stocks with lesser liquidity exposed to little coverage by analysts. The question of whether the use of ETFs increase or decrease the information efficiency of markets was also posed by Glosten et al. (2020). They suggested that the inclusion of ETFs increases information efficiency for low-liquidity stocks.

On the negative side, uninformed traders tend to exit the market in the underlying securities in favor of ETFs, thus depressing liquidity. Furthermore, much ETF activity happens at market close (Shum et al. 2016). Shum et al. noticed that ETFs typically have larger mispricing and wider spreads during end-of-trading, especially on the days of most volatile trading.

If the influence of ETFs is so prominent, can they be a catalyst for extreme events in the market? Some authors, e.g., A. Madhavan, have answered positively (Madhavan and Sobczyk 2019). At least, there is a recognition that new kinds of risks are inherent in the proliferation of ETFs (Pagano et al. 2019). If that is true, can big data analyses and deep learning instruments provide some warning about whether extreme events may be coming? And, what is the direction of the information flow—from ETF to the broader market or vice versa? (Glosten et al. 2020).

To answer this question, we develop a structured methodology, which allows us to determine with some certainty whether ETF illiquidity results from market fluctuations or if it is the other way around. The hypothetical mechanism is as follows: ETF trading initiates the delivery of ETF constituents ("in-kind" transaction) or a cash equivalent ("in-cash" transaction) if the underlying assets are easily tradable (Bhattacharya and O'Hara 2018). If the aggregate volume of ETF executions were small and/or evenly spread in time, this would introduce friction in orderly market execution.

And indeed, there are a few inherent problems. First, there needs to be more clarity as to whether trade imbalance results from the actual economic events in one or more underlying stocks or from the effects of stock aggregation by the ETFs, for instance, changes in the credit risk of the ETF swap counterparty.

Second, because ETF transactions are prominent in hedges, they are highly nonuniform throughout the day (Shum et al. 2016). A company that misses a delivery can wait until the end of the day to close the deal when the market is more stable. This paper does not judge whether ETFs are a "good" or "bad" influence on market liquidity. It strives to clarify the enormous influence ETFs have on the stock market, particularly the direction of information transmission.

According to the foundational models of market microstructure, the principal driver of price changes is the imbalances in supply and demand for a given security (Kyle 1985; Glosten and Milgrom 1985). Currently, imbalance messages can be followed with nanosecond precision. There is probably little value added to further increasing accuracy because signals only propagate a few meters—i.e., the size of the trading room—with already achievable latency (Bartlett and MacCrory 2019). One of the first studies going up to nanosecond granularity was "Price discovery in high resolution" (Hasbrouck 2021). This

wealth of available data creates many problems of its own. The human mind is poorly adapted to rationalize such amounts of data. Furthermore, our senses evolved in 3D space and have difficulty comprehending multidimensional datasets.

One of the principal channels of influence of the ETF market on overall market stability is using ETF shares for shorting and hedging. ETF shares "fail-to-deliver" on settlements for different reasons. Fail-to-deliver could be a signal of actual economic troubles in a company, shorting in expectation of a real price movement by the AP (authorized provider), the analog of the market makers (MM) for stock, or "operational shorting" (Evans et al. 2018). The description of operational shorting in the above-cited paper by Evans, Moussawi, Pagano, and Sedunov is so exhaustive that I provide a somewhat long quote.[2] Before a security is classified as "fail-to-deliver", an imbalance record is created. Usually, the imbalance is cleared by end-of-day trading or the next day before trading hours. The reputation penalty for being cited in imbalances is typically small (Evans et al. 2018).

The reason markets and the SEC do not regulate intraday deliveries with harsher penalties is obscure. We hypothesize that the inherent optionality involved in paying for order flow (PFOF) is partially responsible for this market feature (for PFOF analysis, see, e.g., (Lynch 2022)). If there were substantial fines or a negative reputation associated with the "failure-to-deliver", the PFOF mechanism would suffer disruptions. Consider that an expected negative return for a penalty would overcome the price improvement offered by the wholesaler. Because the non-delivery probability is nonzero, only relatively large price improvements would justify the counterparty risk, and a large volume of trade would miss the wholesaler. A detailed discussion of the issue is outside the scope of this paper.

This work is dedicated to researching methods to rationalize imbalance datasets with nanosecond time stamps. We compress them into two-dimensional "fingerprints", for which a rich array of algorithms developed for analyzing the visual images is already available. The dataset we use is the list of imbalance messages provided by NYSE Arca. "NYSE Arca is the world-leading ETF exchange in terms of volumes and listings. In November 2021, the exchange had a commanding 17.3% of the ETF market share in the US" (Hayes 2022). The special significance of the data for our problem setting is illustrated by the fact that a glitch in the NYSE Arca trading system influenced hundreds of ETF funds in March 2017 (Loder 2017).

Messages in our database have the following types: type "3", type "34", and type "105". Message type 3 is a symbol index mapping (reset) message. Message 34 is a security status message, which can indicate "opening delay", "trading halt", and "no open/no resume status". Finally, message type 105 is an imbalance message. More information about the format and content of the messages and the datasets can be found in Appendix A and (NYSE Technologies 2014).

To make use of the large statistics of the nanosecond time stamps of the 105 messages, we selected them for our analysis. Our choice is justified because the daily stream of 105 messages is in the millions, while 3 and 34 messages are in the tens of thousands.

The number of imbalance messages (type 105) for each trading day is around four million, each comprising 15–20 standardized fields. TAQ NYSE Arca equities—TAQ NYSE imbalance files provide "buy and sell imbalances sent at specified intervals during auctions throughout the trading day for all listed securities" (NYSE Technologies 2014).

## 3. Preprocessing—Formation of the State Variables Database

We selected the following variables: (1) the number of messages per unit time, and price, (2) the dollar imbalance at the exception message, and (3) the remaining imbalance at settlement. The latter is rarely different from zero because a failure to rectify stock imbalances at the close of a trading session indicates a significant failure in market discipline and may entail legal consequences.

Our data can be divided into two unequal datasets: market messages in their totality and ETF-related messages, and the first group encompasses the second. Because of the large volume of the data, we used an algorithmic selection of data for the ETF group. The

messages in the datasets contain the identifier "E" for exchange-traded funds, but in other places, "E" can indicate corporate bonds, but it is too common a letter to filter for it in a text file. Instead, we chose separate data on market participants provided by the NYSE, of which we filtered the names explicitly containing the words "exchange-traded" or "fund". This identification is not 100% accurate because some closed-end mutual funds, which are not ETFs, could have entered our list, but they are expected to be dominated by ETFs.

We were left with 1061 names automatically selected from the messages file. The number of daily events related to our list can be half a million or more, so sorting by hand would be difficult, if possible at all.

We further grouped our data as follows: First, the number of type 105 messages per 100 s. Second, the cumulative imbalance every 100 s in a $12 1/2$ h trading day[3,4]. The number of price bins chosen was approximately equal to the number of time intervals. Dollar imbalances are calculated by the following formula:

$$\$Imb_t = p \cdot (Imb_t - Settle_{4:00}) \tag{1}$$

where $p$ is the last market price, $Imb_t$ is the undelivered number of shares, and $Settle_{4:00}$ is the number of shares unsettled by the end of the trading session, usually at 4:00 p.m.

The 100 s intervals were chosen arbitrarily but intended to have a two-dimensional data tensor processed on a laptop and have sufficiently acceptable statistics. The imbalances are distributed quite irregularly at around 45,000 s and can be visually grouped into the "beginning of the day settlement", "midday settlement", and "end-of-day settlement" (see Figure 1).

As expected, most of the dollar imbalances are small. To avoid data being swamped into a trivial distribution—a gigantic zeroth bin—and a uniformly small right tail, we used a logarithmic transformation for all variables, including time: $\tilde{x}_t = \ln(1 + x_t)$. Unity was added to deal with the zeroes in the database. Nonlinear transformation distorts probability distributions, but given their sharply concentrated and peaked shape, we did not expect it to influence the results too much.

The plot of the summary statistics for a typical day is shown in Figure 2. We observed that the maximum number of imbalance messages created by ETFs for each 100 s during a trading day is about one-eighth of the total number of exceptions (~25,000:200,000) in the market, but the cumulative value of imbalances created by the ETFs is about 60% of the total. This disparity suggests that average imbalances are much higher when ETF shares are involved.

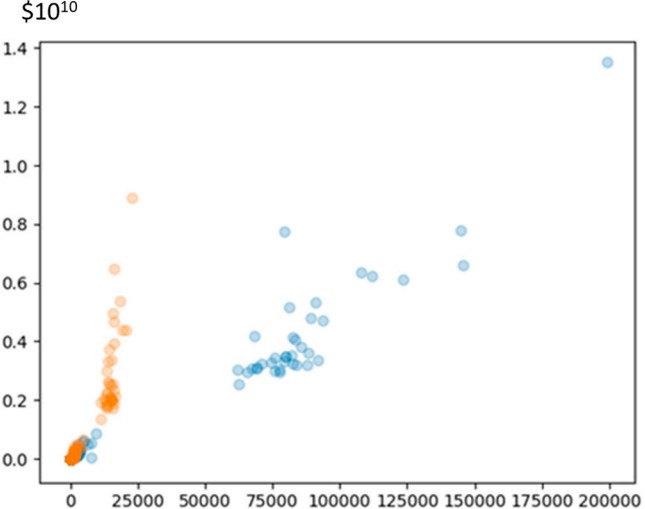

**Figure 2.** Cumulative dollar imbalances as a function of the number of imbalance messages per 100 s period. Orange dots are the transaction imbalances involving only ETF shares, and blue dots are the total market imbalance. We observe that the message rate of ETFs is approximately one-eighth of the total market, but their dollar value exceeds 60% of the cumulative market imbalances.

The final step was making our data amenable to deep learning algorithms, mostly designed to deal with visual data ("fake images"). We further compressed our nonuniform rectangular matrices—in some cases, messaging did not begin at exactly 3:30 a.m., etc. One data file included only NYSE into 96 × 96 squares, which we call "fingerprints" of daily trading.[5] (Figure 3). The fingerprints do not have an obvious interpretation; rather, they take the form of a machine-readable image, like a histogram. The transaction rate was plotted on the vertical scale. We plotted an accompanying dollar value for an imbalance on the horizontal scale. This compression method allows for treating daily imbalances as uniform images, which can be subjected to processing using deep learning algorithms.

Five randomly selected trading days (7–8 October 2019, 9 September 2020, and 4–5 October 2020) produced ten daily samples: one with total market imbalance messages, the other with ETF data only. We constructed five testing and five training samples from them according to the protocol exhibited in Figure 4.

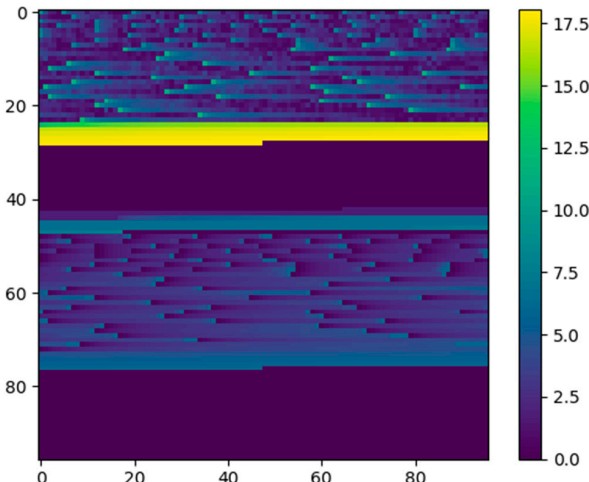

**Figure 3.** Example of the "fingerprint" of a trading day on a natural logarithmic scale. The 105-type message rate and the cumulative dollar amount of imbalances are placed into 96 bins.

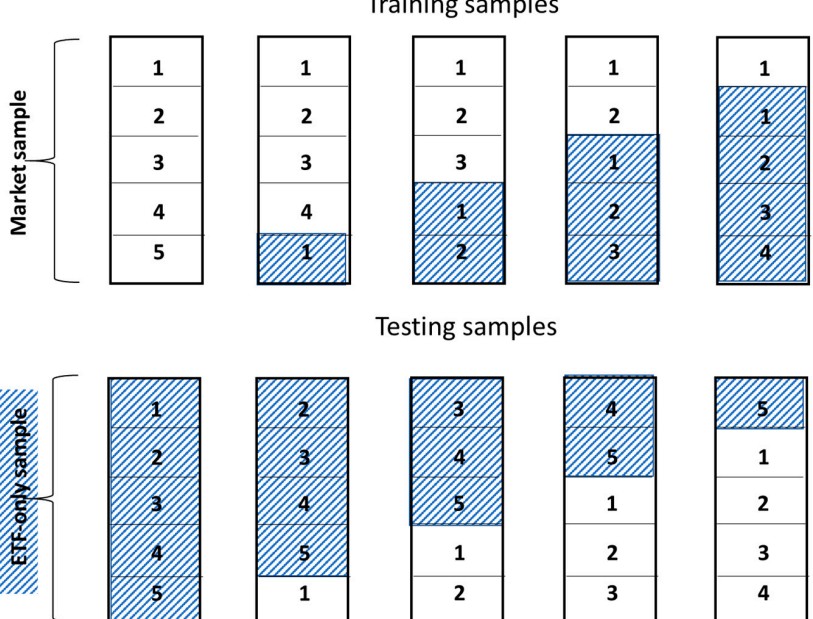

**Figure 4.** Composition of the training and testing samples. Each number indicates a fingerprint of a given day in chronological order (7 October 2019, 8 October 2019, 9 September 2020, 4 October 2021, and 5 October 2021). Note that only the first samples in corresponding columns are mirror images of each other. Blue diagonal pattern indicates ETF-only samples.

A relative proportion of all-market and ETF-only samples, according to Figure 3, is provided in Table 1.

**Table 1.** The proportion of the all-market and ETF-only samples in each simulation.

| Training File | Market:ETF | Test File | Market:ETF |
|:---:|:---:|:---:|:---:|
| tr1 | 100%:0% | tes1 | 0%:100% |
| tr2 | 80%:20% | tes2 | 20%:80% |
| tr3 | 60%:40% | tes3 | 40%:60% |
| tr4 | 40%:60% | tes4 | 60%:40% |
| tr5 | 20%:80% | tes5 | 80%:20% |

## 4. Distances on a State Space

Because of their "black box" nature, the output of neural networks is hard to rationalize. First, the human mind has evolved to analyze two- or three-dimensional images in three-dimensional space. Most humans cannot directly comprehend tensor inputs, intermediate results, and outputs typical for neural networks. Second, the results of neural network analyses are necessarily stochastic and depend on the large number of estimated intrinsic parameters, which are frequently inaccessible, but in any case, too numerous to rationalize. Third, deep learning results can depend on how the training and testing samples are organized, even if they represent identical datasets. All of this can indicate the failure of a deep learning procedure (Brownlee 2021), but it can also show additional information we fail to recognize.[6] Because neural networks are "black boxes", instead of the interpretation of a hundred thousand—in my case trillions of—parameters in the case of Google and Microsoft deep learning networks, one has to design numerical experiments and analyze the output from a deep learning algorithm in its entirety.

To systematize the results, we propose two measures of divergence of images as follows: After the C-GAN generated fake images ("fingerprints") of the session, we considered these images as (1) matrices and (2) nonnormalized probability distributions.

The first approach is to treat arrays as matrices (tensors). We computed the pseudo-metric cosine between the image arrays $X$ and $Y$ according to the following formula:

$$C_{XY} = \frac{\|X + Y\|^2 - \|X - Y\|^2}{4\|X\| \cdot \|Y\|} \tag{2}$$

In the above formula, the norm $\|\cdot\|$ is a Frobenius matrix norm representing each image array. In the first stage, we computed the distance as the average of each twentieth of the last 400 images in the sequence. Because, sometimes, the fake image is an empty list having a zero norm, we modified this formula according to the following prescription:

$$C_{train,fake} = \frac{\|train + fake\|^2 - \|train - fake\|^2}{4\|train\| \cdot \|test\|} \tag{3}$$

$$C_{test,fake} = \frac{\|test + fake\|^2 - \|test - fake\|^2}{4\|train\| \cdot \|test\|}$$

Equation (3) provides answers close to the correct geometric Formula (2), but it does not fail in the case of an empty fake image. The pseudo-metric measure, calculated according to Equation (3), provides a fair picture of the affinity of the fake visual images to the originals (see Figure 5), but it is still unstable with respect to different stochastic realizations of the simulated images.

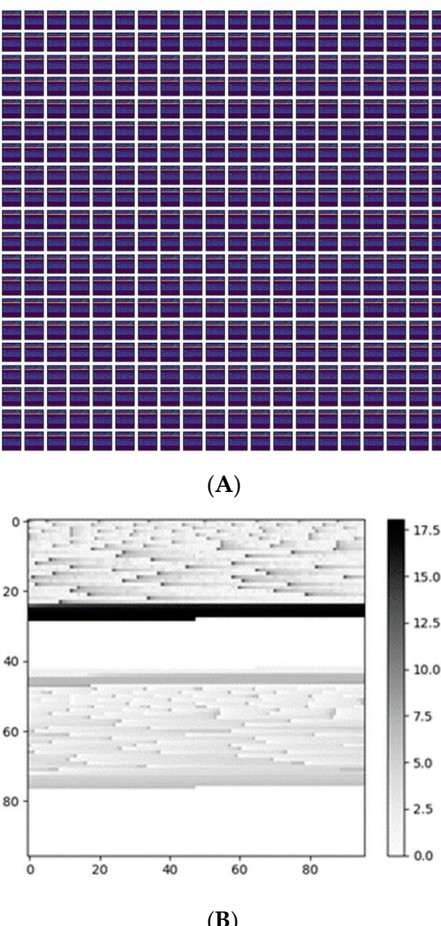

**(A)**

**(B)**

**Figure 5.** The output of C-GAN. (**A**) Set of 400 fake images generated by the generator part of the deep learning network during 1600 epochs. (**B**) An individual fake image (compare Figure 3).

So, we applied a second stage averaging according to the formula for the mutual information:

$$MInfo = \frac{1}{N}\sum_{i=1}^{N} log_2 \left( \frac{C_{train,fake,i}}{C_{test,fake,i}} \right) \tag{4}$$

In Equation (4), $N$ is the number of independent network runs. Note that this formula does not depend on whether we use a "geometrically correct" Equation (2) or a computationally convenient Equation (3). Unlike separate $C_{train,fake}$, and $C_{test,fake}$ norms, which may vary widely between consecutive runs of the C-GAN, their ratio is reasonably stable for a given training and testing sample. Furthermore, if one exchanges training and test samples, the argument of the summation of Equation (4) only changes its sign.

The second option is to treat output arrays as quasiprobability distributions. We used Kullback–Leibler (KL) divergences $D_{KL}(P||Q)$ between the two distributions (Burnham and Anderson 2002). As is well known, Kullback–Leibler divergence is not a real metric and is asymmetrical with respect to its arguments. The intuitive meaning of $D_{KL}(P||Q)$ is an information gain achieved when one replaces distribution $Q$ (usually meaning sample) with distribution $P$ (usually meaning model). In our context, distribution $P$ is a training or test dataset, and distribution(s) $Q$ are fakes generated using the neural net. A final information criterion is:

$$r = \frac{D_{KL}\left( X_{test} \middle\| X_{fake} \right)}{D_{KL}\left( X_{train} \middle\| X_{fake} \right)} \tag{5}$$

The values $r > 1$ suggest that the test file is difficult to reproduce by training the net. The values $r < 1$ indicate that reproducing the dataset—the neural network does not know anything about trading, which generates observable data—is relatively easy.

There are also several variations for the design of the samples we analyzed. In the first preliminary design, we compared data of the entire market, which includes ETFs, against the data files, which contain only our sample(s) of ETFs. In that case, we expected a positive sign of mutual information when we trained the network using the entire market file—because the information from the ETFs is already contained in a training file. We display these results in the next section.

In the second design, we separated data files into single-issue and ETF stocks. The positive sign of the mutual information must appear if the training occurs on the more informative information subset of the two. We identified a more informative data file as causally determining a less informative data file.

Despite the complexity of the described procedure, the intuition behind it is quite simple. A convolutional generative adversarial network generates many fake images inspired by a training sample. These images are compared with a test sample. If the fake images perfectly fit both the training and test samples, the mutual information between them is exactly zero. Vice versa, very divergent training and test distributions suggest much additional information, which must be known to reproduce a dataset.

Positive mutual information or a higher correlation between a fake distribution and training one than between a fake and test distribution means that a training file is easier to fake than the test. On the contrary, small or negative mutual information suggests that the C-GAN's operation produces fakes, which are relatively easily distinguishable from the training file.

## 5. Preliminary Results of C-GAN Analysis

The results of the fingerprint analysis using the C-GAN are displayed in Tables 2 and 3. In that analysis, we used our vocabulary of 1060 funds automatically selected by words in their name. For the robustness of this choice, we also tested the list of 1753 US ETF funds in the database: https://www.etf.com/channels/us-etfs (accessed on 20 May 2021).[7] We display five runs of the C-GAN network with the construction of samples according to the first line from Table 1. The comparison with the first cell of Table 3 suggests that with this second sample of ETFs, mutual information is only strengthened in the direction of "training by the overall market" and "testing by the ETF-only file".

Each cell in Table 2 is the binary logarithm of the ratio of the distances of the generator-devised fingerprint between training and testing samples, respectively. Table 3 shows that the mutual information generally decreases with a diminishing fraction of market samples and increasing ETF samples. The fraction of the market vs. ETF samples in the testing samples demonstrates no visible tendency, with one exception. When the testing file is almost entirely composed of the market samples, mutual information becomes zero irrespective of the training samples (Tables 2 and 3).[8]

The column and row averages are provided in Table 3. Testing sample 2 is an outlier. We tentatively attribute it to one of the day's data being exceptional. And indeed, the file for 9 September 2020 contained, probably, only stocks listed on NYSE, not all traded stocks. All averages are positive. This suggests that all-market files are easier approximated by the generator-produced fakes than the ETF-only files. We consider this evidence that ETF-only files have more distinguishing features than the all-market files and, consequently, are more distant from fakes than the training files, i.e., if one used ETF data as a training set, it was relatively easy to train the network to have a relatively high correlation of fakes with all-market samples. On the contrary, the all-market training set was insufficient to train the network to distinguish fakes from the original ETF data.

In a further robustness check, we tested another sample of 1753 ETFs selected using an online database in the same setup. The results are similar to our ad-hoc tests of the 1060 computer-selected funds (Table 4).

**Table 2.** The results of measuring MInfo (Equations (2) and (3)) between the 20 fake images created using the generator and the training and test images. C-GAN was run for 1600 epochs, and the fake images were taken uniformly from the last 400 images.

|  | tr1 | | tr2 | | tr3 | | tr4 | | tr5 | |
|---|---|---|---|---|---|---|---|---|---|---|
| tes1 | 0.7855 | 0.6538 | 0.1511 | 0.7556 | −0.4958 | −0.0203 | 0.2263 | −0.1734 | −0.6959 | 0.8837 |
| tes2 | 0.5147 | - | 1.0126 | - | −0.2086 | - | 0.5735 | - | −1.4379 | - |
| tes3 | 2.5042 | 2.6258 | 2.2150 | 2.5771 | 2.2438 | 2.0920 | 2.5458 | 2.3841 | 2.0923 | 2.2976 |
| tes4 | 0.3850 | 0.3785 | 0.4250 | 0.7784 | 0.7484 | 0.0000 | 0.1193 | 0.9425 | −0.0687 | −0.4286 |
| tes5 | 0.0000 | - | 0.0000 | - | 0.0000 | - | 0.0000 | - | 0.0000 | - |

**Table 3.** Mutual information (Equation (3)) for the training and test samples.

| Training File | Test File Average | Test File | Training File Average |
|---|---|---|---|
| tr1 | 0.9809 | tes1 | 0.2071 |
| tr2 | 0.9894 | tes2 | 0.0909 |
| tr3 | 0.5449 | tes3 | 2.3578 |
| tr4 | 0.8273 | tes4 | 0.3280 |
| tr5 | 0.3303 | tes5 | 0 |

**Table 4.** Mutual information for the selection of 1753 ETFs from etf.com. The arrow shows the direction from the training to the test file in the C-GAN network. SU denotes "NYSE Stock Universe" data, and "ETF" denotes ETF-only data. The arrow indicates the direction from training to test files.

| Runs | MI SU→ETF | MI ETF→SU |
|---|---|---|
| 1 | 0.8605 | 0.2059 |
| 2 | 1.0595 | −1.0925 |
| 3 | 0.8900 | 0.2510 |
| 4 | 0.6670 | −1.0926 |
| 5 | 1.0403 | 0.1261 |
| Average | 0.9035 | −0.3204 |
| Std. dev. | 0.1589 | 0.7063 |

## 6. The Test of Single-Issue Stocks against ETF Data

We conducted a test wherein we excluded the data for 1753 ETFs from the market data. The results of the test are displayed in Table 5.

We observed that one can teach the network by feeding it with ETF-only data. Our network successfully interpolated single-issue stock data by the "learned fakes" but not vice versa. We tentatively make the case that ETF trading provides more information for traders in single-issue stocks, and henceforth, the direction of causality is from ETFs to single issues. A possible economic explanation for this phenomenon is provided in Appendix B.

The treatment of the neural network outputs as probability distributions allows for using another measure on the state space, namely Kullback–Leibler distance (see Section 4). The results of computing the parameter $r$ from Equation (5) are shown in Table 6.[9] The layout of this table reflects the difficulty of expressing relations between tensors in a human-readable format.

In each of the cells, except one—the upper second from the left—a lower asymmetry between the proportions of ETF and single-issue samples in test and training files corresponds to a higher value of $r$. Average values of $r$ in the cells are the largest for a low (25% and 50%) fraction of the ETF samples in the test file. Henceforth, a GAN network can falsify the single-issue data more successfully.

**Table 5.** Mutual information for the selection of 1753 ETFs from etf.com. The arrow shows the direction from the training to the test file in the C-GAN network. SI denotes "Single-issue stock" data, and "ETF" denotes ETF-only data. The arrow indicates the direction from training to test files. We observed that the ETF-trained network successfully teaches SI-enabled data, while the obverse is impossible.

| Runs | MI ETF$\rightarrow$SI | MI SI$\rightarrow$ETF |
|---|---|---|
| 1 | 0.6857 | −0.0495 |
| 2 | 0.9169 | 0.2002 |
| 3 | 0.9599 | −0.0520 |
| 4 | 0.9822 | −0.0670 |
| 5 | 1.0789 | 0.0675 |
| Average | 0.9247 | 0.0199 |
| Std. dev. | 0.1462 | 0.1143 |

**Table 6.** The ratio of Kullback–Leibler distances $r$ (Equation (5)) as a function of the share of ETF samples in test data (1) and the difference between the proportion of ETF and single-issue samples in the test and training files, respectively (2). Averages in each group are indicated by (3). In each cell except one, lower asymmetry entails a higher $r$ index. Color coding is added for visibility.

| Averages (Rows) | ETF Fraction | ETF Fraction Difference between Training and Test Sets | | | | | | | | ETF Fraction | Averages (Rows) |
|---|---|---|---|---|---|---|---|---|---|---|---|
| | | 100% | 50% | 0% | −50% | 100% | 50% | 0% | −50% | | |
| 2.252 | 75% | | 4.251 | | 0.802 | | 9.865 | | 3.15 | 25% | 4.455 |
| | 75% | 2.665 | | 2.023 | | 1.983 | | 2.823 | | 25% | |
| 1.892 | 100% | | 2.330 | | 1.440 | | 7.787 | | 41.549 | 50% | 13.142 |
| | 100% | 2.031 | | 1.331 | | 1.927 | | 1.305 | | 50% | |
| Averages (columns) | | 2.819 | | 1.399 | | 5.391 | | 12.207 | | | Averages (columns) |

## 7. Conclusions

In this paper, I presented a new econometric methodology to investigate the causality of the financial time series. In variance to original Granger causality, this methodology does not rely on any explicit model of the stochastic process by which the input data were generated. It is preferable to nonparametric Granger causality techniques in the case of extra-large or multidimensional datasets because it does not rely on the computation of correlations between multiple subsets of the original data.

The proposed method was applied to solve an important question: whether individual stocks or ETFs drive the liquidity of markets. I chose the information content of the number of imbalances to measure liquidity. The latter indicates the inability to instantly fill the quote at a given price and the dollar value of incomplete transactions. The information content was measured as a pseudodistance between the time series in a two-dimensional state space (the number of a price bucket and its dollar imbalance).

The preliminary answer is that both the rate of imbalance arrivals and the dollar value of resulting imbalances of the ETFs are more informative—in the sense of finer features nonreproducible by fakes—than the individual stocks with ETFs counted as separate stocks. Higher information content of ETF imbalances is not surprising. Indeed, the imbalance messages produced by 1000+ ETFs constitute about one-eighth of the totality of exceptions in the entire database on average, but the dollar value of their imbalances is about two-thirds of the entire dollar value of the market imbalance. Theoretically, this is not surprising because, as pointed out by (Shum et al. 2016) and (Evans et al. 2018), ETF securities are used for hedging much more frequently than individual stocks. An explanation of this phenomenon based on extant economic theory is provided in Appendix B.

**Funding:** This research was prepared without using external funding.

**Data Availability Statement:** The author provides all research data and the source code on request.

**Acknowledgments:** The author thanks the participants of the World Finance Conference (Braga, Portugal, discussant: Penka Henkanen), FEM-2022 (ESSCA, Paris, discussant: John Beirne), the Risk Management Society Conference (Bari, Italy, discussant: Linda Allen), and especially the Asia Meeting of Econometric Society 2023 (Beijing, China) discussant Claudia Moise including the reference for her working paper.

**Conflicts of Interest:** The author declares no conflict of interest.

**Appendix A**

# Description of the TAQ ARCA messages

### 2.11 SYMBOL INDEX MAPPING MESSAGE (MSG TYPE '3')

| FIELD NAME | FIELD ORDER | FORMAT | DESCRIPTION |
|---|---|---|---|
| MsgType | 1 | Numeric | This field identifies the type of message. '3' – Sequence Number Reset message |
| SequenceNumber | 2 | Numeric | Message sequence number by channel. Must derive sequence number based on leading sequence number for each packet. |
| Symbol | 3 | NYSE Symbology | This field displays the symbol in NYSE symbology |
| SymbolIndex | 4 | Numeric | This field identifies the numerical representation of the NYSE symbol. This field is unique for products within each respective market and cannot be used to cross reference a security between markets. |
| Market ID | 5 | Numeric | ID of the Originating Market: ■ '1' - NYSE Cash |

TAQ NYSE Arca Integrated Feed /V1.8

20

### 2.15 SECURITY STATUS MESSAGE (MSG TYPE '34')

| FIELD NAME | FIELD ORDER | FORMAT | DESCRIPTION |
|---|---|---|---|
| MsgType | 1 | Numeric | This field identifies the type of message. '34' – Security Status Message |
| SequenceNumber | 2 | Numeric | Message sequence number by channel. Must derive sequence number based on leading sequence number for each packet. |
| SourceTime | 3 | HH:MM:SS.nnnnnn | This field specifies the time when the msg was generated in the order book. The number represents the number of seconds at microsecond accuracy in UTC time (since EPOCH) |
| Symbol | 4 | NYSE Symbology | This field displays the symbol in NYSE symbology |
| SymbolSeqNum | 5 | Numeric | This field contains the symbol sequence number |
| Security Status | 6 | ASCII | The following are Halt Status Codes: ■ '3' - Opening Delay |

TAQ NYSE Arca Integrated Feed /V1.8

25

## 2.9 IMBALANCE MESSAGE (MSG TYPE '105')

| FIELD NAME | FIELD ORDER | FORMAT | DESCRIPTION |
|---|---|---|---|
| Msg Type | 1 | Numeric | This field identifies the type of message. |

TAQ NYSE Arca Integrated Feed /V1.8

| FIELD NAME | FIELD ORDER | FORMAT | DESCRIPTION |
|---|---|---|---|
| | | | '105' – Imbalance Message |
| SequenceNumber | 2 | Numeric | Message sequence number by channel. Must derive sequence number based on leading sequence number for each packet. |
| SourceTime | 3 | HH:MM:SS.nnnnnn | This field specifies the time when the msg was generated in the order book. The number represents the number of seconds at microsecond accuracy in UTC time (since EPOCH) |
| Symbol | 4 | NYSE Symbology | This field displays the symbol in NYSE symbology |
| SymbolSeqNum | 5 | Numeric | This field contains the symbol sequence number |
| ReferencePrice | 6 | Numeric | The Reference Price is the Last Sale if the last sale is at or between the current best quote. Otherwise the Reference Price is the Bid Price if last sale is lower than Bid price, or the Offer price if last sale is higher than Offer price. |
| PairedQty | 7 | Numeric | This field contains the paired off quantity at the reference price point |
| TotalImbalanceQty | 8 | Numeric | This field contains the total imbalance quantity at the reference price point |
| MarketImbalanceQty | 9 | Numeric | This field indicates the total market order imbalance at the reference price |
| AuctionTime | 10 | hh:mm | Projected Auction Time |
| AuctionType | 11 | Alpha | ■ 'O' – Open (4am) Arca Only<br>■ 'M' – Market (9:30am)<br>■ 'H' - Halt<br>■ 'C' – Closing<br>■ 'R' – Regulatory Imbalance<br>**Note:** For the NYSE/MKT, the opening imbalance will have an "M" Auction Type |

# Message type '105'

| FIELD NAME | FIELD ORDER | FORMAT | DESCRIPTION |
|---|---|---|---|
| ImbalanceSide | 12 | Alpha | This field indicates the side of the imbalance Buy/sell. Valid Values: <br> ■ 'B' – Buy <br> ■ 'S' – Sell <br> ■ ' ' – No imbalance <br><br> Note: This field is a future enhancement for NYSE Arca and will have a '0' value until such time. |
| ContinuousBook ClearingPrice | 13 | Numeric | The Continuous Book Clearing Price is defined as the price closest to last sale where imbalance is zero. <br><br> If a Book Clearing Price is not reached, the Clearing Price, a zero will be published in the Book Clearing Price Field <br><br> Note: This field is a future enhancement for NYSE Arca and will have a '0' value until such time. |
| ClosingOnly ClearingPrice | 14 | Numeric | This field contains the indicative price against closing only order only <br><br> Note: This field is a future enhancement for NYSE Arca and will have a '0' value until such time. |
| SSRFilingPrice | 15 | Numeric | This field contains the SSR Filing Price. This price is the price at which Sell Short interest will be filed in the matching in the event a Sell Short Restriction is in effect for the security. <br><br> Note: The SSR Filing price is based on the National Best Bid at 9:30am. This price remains static after the SSR Filing price has been determined. <br><br> Note: This field is a future enhancement for NYSE Arca and will have a '0' value until such time. |

### The format of TAQ ARCA messages with a nanosecond time stamp

34,10,00:24:58.796044288,CBO,1,P,~,,,,,,~,P

3,11,BANC,1,51,N,C,100,11.47,0,0,N,.0001,1

105,273294,09:29:34.061214976,UHT,33,66.21,170,1141,0,
0930,M,B,67.67,0,0,0,0,0,0,0,0,0, ,

**Appendix B**

The problem of ETFs as potential market drivers was explored by Semyon Melamed (Melamed 2015), who provided a dynamic equilibrium theory of the interaction of ETFs and the general market. In particular, he proved that if approved participants (AP, analog of market makers) act as arbitrageurs, correlations in a broad market increase. Moreover, stocks included in physical ETFs influence correlations more than stocks in synthetic-only ETFs. This effect can be amplified by less liquid securities in some of the physical ETF portfolios (Marta and Riva 2022). These securities are inevitably present in the ETF tied to MSCI World, Russell 2000, and other popular indexes. Then, ETF participants can emulate not the entire portfolio but a representative subset (see also (Koont et al. 2022)). However, Melamed's theory is difficult to use for practical estimation because of many unobservable parameters. A more parsimonious model was proposed by Pan and Zeng (2017). Technically, their treatment calls a risky illiquid asset a "bond", but for intraday trading, it makes no difference with stock. In particular, we can use Theorem 6.1 to elucidate the information content of the action of APs (Koont et al. 2022; Melamed 2015; Pan and Zeng 2017) in ETFs. One of the principal results of Pan and Zeng was Equation (A2) for the optimal number of shares issued $z^*$. If we assume a high correlation between ETF and illiquid securities lying in the foundation of its portfolio, it is convenient to make a substitution for the correlation factor $\rho \to 1 - \epsilon \le 1$. This assumption is certainly true when applied to the entire ETF universe. Then, we can rewrite Equation (A2) uisng a slightly different notation:

$$z^* = \frac{(\lambda + \theta\sigma^2)(\pi + c_L) + \lambda\epsilon(\pi - c_E)}{\lambda(\lambda + 2\mu) + 2\epsilon\sigma^2(\lambda + \mu)} \tag{A1}$$

Here, in Equation (1), $\lambda$ is the Kyle–Amihud illiquidity factor, $\mu$ is the flotation cost of the ETF share, $\theta$ is the risk avoidance factor, $\sigma$ is the volatility, $c_L = c_B + \lambda x_{B-} - c_E \ge 0$ is the difference between price movement of the illiquid ($c_B$) and liquid asset ($c_E$) for the customer order $x_{B-}$.

The current study quantitatively measures the enhanced informational content of the ETF imbalances by Equation (A1). For instance, for the pair tes1–tr1 (Table 2), this suggests that per dollar imbalance basis, the ETF transaction is $2^{0.7855} = 1.724$ times more informative than an action of the entire market, ETFs included.

The measure of the informativeness of AP trades is hardto design. Yet, if we assume that it is some version of entropy, i.e., a convex functional on the optimal number of newly issued shares, then the first-order condition will be

$$\frac{\delta S(z)}{\delta z}\Big|_{z=z^*}\delta z^* = 0$$

An approximate formula for the FOC becomes

$$\frac{\pi - c_E}{\pi + c_L} = \frac{2\theta\sigma^2(\lambda + \mu + \theta\sigma^2)}{\lambda(\lambda + 2\mu)} \tag{A2}$$

The price mismatch near the equilibrium has an order of $\frac{2\theta\sigma^2}{\lambda}$, i.e., the volatility is greatly amplified if the risk avoidance factor is nonzero and the Kyle–Amihud constant is

small.[10] Given the same level of risk aversion, higher liquidity indicates higher information content. All of the above supports the conclusion of higher information brought upon by the AP orders.

## Notes

1    Hasbrouck, https://www.youtube.com/watch?v=EZCgW1mFRP8 2010 (accessed on 20 May 2021).

2    "One important objective of APs in the primary ETF market is to harvest the difference between ETF market price and its NAV . . . As demand for the ETF grows from investors in the secondary market, the ETF's market price should increase [Increasing the possibility of market arbitrage—P. L.] However . . . selling ETF shares and buying the underlying basket/creating the ETF shares are not necessarily instantaneous. The AP sells the new ETF shares to satisfy bullish order imbalances but can opt to delay the physical share creation until a future date. By selling ETF shares that have not yet been created, the AP incurs a short position for operational reasons . . . that we hereafter call an "operational short" position." The paper (Evans et al. 2018) also lists "directional shorting", i.e., speculation on the changing market price as a reason for "fail-to-deliver.".

3    Our messages begin at 3:30 a.m. and end at 4 p.m., 45,000 s in total, usually, but not always, 449,100 s intervals starting with zero. Each time interval contains ~9000 messages on average. Yet, the highest message rate in each price bin can be more than twenty times as high.

4    The price bin methodology is reminiscent of the VPIN measure of Easley et al. (2013). We experimented with linear as well as the logarithmic scale of our data. In this paper, we use a logarithmic scale.

5    Each fingerprint contains 9216 pixels. We compress our ~400 MB daily database into ~200 K text file, a compression of 2000 times.

6    The output from C-GAN indicates a deep learning failure called "mode collapse" (Brownlee 2021). Yet, the look-alike of the fake images remains excellent.

7    We used the data from only four days in our sampling because, for unknown reasons, one of the TAQ ARCA files has no overlap with the second ETF list.

8    We must issue a caution that most applications of the GAN networks suffer from overfitting and mode drop (Yazici et al. 2020). Visual inspection of the losses by the critic and the generator suggests that it can take place but, currently, we can do nothing about it. Variation in the number of epochs, discrimination tolerance and other standard remedies do not change the qualitative picture.

9    Data in Table 6 results from a single run of the C-GAN network.

10   The small Kyle-Amihud constant is a reasonable assumption given the relatively high liquidity of the NYSE-traded shares.

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
