# Peer review of "A New Entropic Measure for the Causality of the Financial Time Series"

_jrfm, doi:10.3390/jrfm16070338_

Round 1
Reviewer 1 Report
1. Abbreviation in the abstract should be used only after defining it.
2. Youtube link can be given as footnote.
3. Further, what is the authenticity of such videos? Can authors use any research work instead?
4. Introduction should clearly be written highlighting the context, motivation, and also emphasize the contribution of the work.
5. Numbering of sections are wrong.
6. Citations are not given in appropriate format.
7. The paper needs through English editing for academic writing. Sentence formation, language flow needs a significant improvement.
8. The paper is not reader friendly. Authors should improve structuring of the paper.
9. How this work is different than the other similar work. Not much of discussion is there. Authors needs to discuss and compare this work with others to bring the logical importance of this work.
1. Tables, figures are not formatted properly.
The paper needs through English editing for academic writing. Sentence formation, language flow needs a significant improvement.
Author Response
Thank you for the detailed review. Attached is my response,
Peter

Reviewer 2 Report
Please, check the attached file.

Author Response
Thank you for thoughtful review. My response to your comments is attached,
Peter

Round 2
Reviewer 1 Report
I think the authors have addressed the issues raised in the review comments.
Reviewer 2 Report
The author applied all comments and I see the revised copy has been accepted for publication in the journal.